# Identifying the Enablers and Barriers to Advance Nurse Prescribing of Medication in Spain According to Experts’ Views: A Delphi Study

**DOI:** 10.3390/ijerph20064681

**Published:** 2023-03-07

**Authors:** Francisco Javier Gomis-Jimeno, Manuel Lillo-Crespo

**Affiliations:** 1Department of Nursing, Faculty of Health Sciences, University of Alicante, 03690 Alicante, Spain; 2HLA Vistahermosa Hospital, 03015 Alicante, Spain

**Keywords:** nurse prescribing, pharmaceutical care, legislation, Delphi study

## Abstract

Nurses play an important role in pharmaceutical care worldwide in detecting clinical changes, communicating and discussing pharmacotherapy with patients, their families, and other healthcare professionals, proposing and implementing drug-related interventions, and ensuring the monitoring of patients and their medication regimens, among others. However, there is no global consensus across countries regarding the prescribing of medication by nurses. In Spain, for example, this topic is currently in transition since the approval of the Royal Decree 1302/2018 of October 22nd, which regulates the indication, use, and authorization for dispensing human-use medication by nurses. Our study aims to identify the enablers and barriers to advancing the nurse prescribing of medication in Spain through the views of experts in the field and according to the latest Royal Decree approved and the steps taken by the different Spanish autonomous communities. A modified qualitative Delphi study with three iterations was performed online through the perspectives of experts from the field of healthcare education, research, practice, management, and policy. Data extracted from the literature review were used to formulate the open-ended questions utilized in the three rounds. The experts involved (n = 15) belonged to different Spanish regions where the Royal Decree is being implemented with different speeds, and had distinct backgrounds and experiences. Our results highlight the importance of prospectively developing additional protocols based on chronic diseases as well as scaling up towards independent nursing prescription, the inclusion of a joint multidisciplinary pharmaceutical care model, the controversial role played by national nursing councils and boards, the variability in the speed of implementation among the autonomous communities, and the lack of nursing training in the field of medication prescription.

## 1. Introduction

Nurse prescribing of medication (NPM) in Spain has been defined according to the Spanish General Council of Nurses in 2007 as the capacity of the nurse for the management, evaluation and provision of nursing care, as well as the indication of different materials, product-devices, and medication aimed at satisfying the health needs of the user and the population, guided by professional criteria, supported by clinical judgment, and managed in the form of care [1]. This term is understood in Spain and the rest of Europe under the umbrella of another one known as pharmaceutical care (PC), which is defined as the responsible provision of drug therapy for the purpose of achieving definite outcomes that improve the patients’ quality of life [2]. Obviously, PC includes not only nurses but also other health professionals such as physicians and pharmacists. However, European healthcare professionals have differences in the implementation and management of the tasks regarding PC such as prescribing, dispensing, delivering, administering medication, providing patient education, and monitoring and evaluating the effectiveness and efficacy of the medications, sometimes with distinct and sometimes with overlapping roles, depending on the country and sometimes even depending on the region [3,4,5,6,7]. To overcome this situation and tend to consensus, it is worth noting the initiatives of the Nuphac European Network, aimed at delineating the functions of nurses in interprofessional PC, showing the ideal roles of nursing, and, in turn, creating the opportunity to translate them into study programs oriented to such competencies and adjusted to the expectations of the labor market [8]. In addition, it is worth pointing out the development of other European Commission funded projects such as Demophac Project, a pioneer in the development of a Pan-European model of PC that carried out numerous studies with the participation of different European countries, being able to implement these skills through the collaboration of nurses, pharmacists and doctors. Prior to Demophac, the Eupron project also highlighted nurses’ roles and responsibilities regarding PC in Europe such as providing patient education and information, monitoring medication adherence, adverse and therapeutic effects, and prescribing medicines [9].

The regulations on NPM in Spain have evolved since the first Law 44/2003 of 21st November 2003 was created concerning the prescription of care and comprising comprehensive health care services, assuming cooperation, a multidisciplinary approach, the integration of processes and care continuity, and avoiding the division between the care processes performed by different specialists. Later on 6th May 2005, the Royal Decree 450/2005 on nursing specialties was created, stressing the importance of nursing prescription and promoting specialized advanced practice nursing roles in Spain; subsequently, in 2006, a modification of the previous law was made, giving rise to Law 29/2006 of 26th July, which concerned guarantees and the rational use of medicines and health products through which a nurse had the autonomous capacity to indicate, use and authorize medicines and health products not subject to medical prescription through the corresponding dispensing authorization and based on clinical practice guidelines [10]. However, the term “prescription” has never been officially used when this activity was carried out by a nurse, since according to the Ministry of Health in 2015: “physicians, dentists and chiropodists, within the scope of their respective competences, are the only professionals’ prescribers, that is, with the right to prescribe medicines subject to medical prescription”, though in Spanish society everyone refers to it as the “nurse prescription of medication” (NPM) [11].

In line with this, it should be noted that the Andalusian autonomous community, as the pioneer in this field in Spain, published the BOJA Decree 307/2009 of July 21^st^, defining the performance of nurses in the field of pharmaceutical provision within the Public Health System of Andalusia. This decree allowed nurses from the Andalusian Public Health System to use and recommend medicines not subject to medical prescription, as well as the indication or prescription of other health products funded by the Andalusian Public Health System, establishing the medicines and health products that the nurses could indicate [12]. However, all of these advances in NPM suffered a great setback with the publication of RD 954/2015 on 23rd December 2015, which regulated the indication, use, and authorization for dispensing human-use medication and products by nurses [13]. Such situation has resulted in crucial changes to the nursing profession, to care dynamics, and to the safety of the patients at all care levels. Since then, various modifications have arisen over the years with regard to the competence of the Nursing Prescription, which is currently in force thanks to the creation of the last Royal Decree 1302/2018 of 22 October, which modifies Royal Decree 954/2015 of 23 October, that nowadays regulates the indication, use, and authorization for dispensing human-use medication and products by nurses, although it will be conditioned, in the terms established by each protocol and clinical practice guideline, both in the field of general care and specialized care, approved by the Permanent Pharmacy Commission of the Interterritorial Council of the National Health System in Spain, and validated by the General Directorate of Public Health, Quality and Innovation of the Ministry of Health, Consumption and Social Welfare, as indicated by the BOE in number 256, of 23 October, 2018 on pages 102,636 to 102,643 [14].

Moreover, in the four-year undergraduate nursing program at Spanish universities, only one course of at least 6 ECTS in Pharmacology is generally taught during the second academic year. In 2017, a study explored the training related to Pharmacology in all of the Spanish nursing undergraduate programs [15], highlighting that the current nursing education is supposed to provide the appropriate knowledge and skills to prescribe drugs and health products without the need to take additional training, although other studies have stressed nurses’ lack of self-confidence with regard to such skill [16]. That investigation was conducted with different health professionals involved in PC in Spain, including physicians, pharmacists and nurses, and concluded that there is a lack of clarity about what professionals involved in this field perceive about NPM in Spain in present times. Therefore, the aim of this study was to identify the enablers and barriers according to experts in the field of NPM, taking into account the current situation in Spain regarding the implementation of the Royal Decree.

## 2. Materials and Methods

### 2.1. Study Design

A qualitative modified Delphi study was conducted to explore the views of experts in the field throughout three rounds of open-ended questions that were answered via email during the last months of 2022. Delphi studies consist of different phases of data collection and analysis, following an iterative process designed to reach a consensus on the relevant aspects explored [17]. In our case, this involved the facilitators and barriers to advancing NPM in Spain. The language used for all communications in our study was Spanish. At each round, the principal researcher explained (through email) the process to be followed by the participants, and a template file was attached to the email with the questions to be answered, round by round, by the experts. Each of the documents with the questions and answers was identified with an ID code corresponding with the experts in order to guarantee participant privacy and confidentiality. Data extracted from the literature review, including both scientific publications and grey literature, were used to design and formulate the open-ended questions utilized in the three rounds.

### 2.2. Sampling and Recruitment: The Experts Panel

All of the experts came from different areas such as policy (from national boards of nursing and councils of nurses), education (from higher education institution), research (with a background of publications regarding the topic), management (from national clinical and community contexts), and practice (from different specialties), and had experience in previous training or studies on NPM. The selection of a sample size of 25 respondents as an expert panel was considered to be sufficient for the purpose of the study [18], and all of them accepted participation in the beginning, though only 15 experts continued until the end of the study. The 15 participants belonged to different autonomous communities in Spain: the Valencian community (seven experts), Andalusia (two experts), Castilla La-Mancha (one expert), Galicia (two experts), Madrid (one expert), Basque Country (one expert), and Catalonia (one expert). In Round One, the 25 experts that agreed to participate at the beginning of the study were sent the email with instructions, although after the data collection only 18 responses were obtained. In Round Two, 15 responses were obtained. Later, in Round Three, responses were obtained from 15 individuals. All of them were previously contacted individually through their public corporate emails and were informed about the study’s objectives. After deciding whether to participate, they provided informed consent. None of them rejected the invitation in the beginning though some refused later to continue participating in the study. Experts provided a variety of perceptions on the contents proposed, taking into account that each of their autonomous communities were at a different historical moment in terms of the NPM implementation. The experts did not know the identity of the others on the panel.

In addition, with Round three, the sociodemographic, professional, and academic characteristics of each expert were collected (see Table 1 and Table 2). Those characteristics were selected with the aim to guarantee experts’ privacy. In this way, data was obtained that made the results of the study relevant and specific.

### 2.3. Data Collection and Analysis

Our Delphi study was an iterative process, and therefore data collection and analysis for each round will be described sequentially.

#### 2.3.1. Round One: Identifying initially the Potential Enablers and Barriers of Medication in Spain

The aim of this first iteration was the identification of the potential barrier factors and facilitating factors, and each of the questions requested from the experts at least three examples of each factor. The email sent also contained background information about the study objective and user instructions. In addition, two documents were attached: an informed consent form for participation and the template with the questions corresponding to that round, specifying that the answers should be sent back in 15 days. This email was sent in September 2022. See in Figure 1 the questions corresponding to Round One:

#### 2.3.2. Round Two: Implementation of NPM in the Health System and the Nurses’ Perspectives

Round Two was constructed based on the results of the first round, with the aim of determining the implementation of the NPM in the Spanish health system, the possible discrepancies between health professionals regarding such implementation, and the involvement of the nurses, associations, societies and other nursing professional organizations. The experts received another email with the results of Round One, and a new file was attached to complete the questions of Round Two. This email was sent in October 2022. See in Figure 2 the questions corresponding to Round Two:

#### 2.3.3. Round Three: Developing New Clinical Practice Guidelines, Protocols, Specific Training and the Certification for Nurses

From the results of Round Two, many of the topics related to NPM in Spain were found to be useful for generating the third questionnaire, whose aims focused on current clinical practice protocols or guidelines, with special emphasis on possible improvements for the creation of future specialized ones. Moreover, it was decided to complementary ask them about the training available and the accreditation process required for nurses in Spain. Furthermore, a new file was also attached to be completed by the experts, asking about their sociodemographic data, including their gender, autonomous community and province, professional position, professional experience in years, academic level, as well as a question regarding their areas of expertise, research, or projects in which they have participated in relation to NPM. This email was sent in November 2022. See in Figure 3 the questions corresponding to Round Three:

Finally, the experts were acknowledged for their participation in the study, and they were informed that they would receive an email with the final results of the study once they were available.

## 3. Results

The qualitative content analysis conducted involved a process designed to condense raw data into themes after each round that is based on valid inference and interpretation. For this process, inductive reasoning was used, by which themes emerged from the data after each round through the researchers’ careful examination and constant comparison. Even though the experts came from different backgrounds, there was a consensus in most of the responses.

### 3.1. First Iteration 

In Round One, all of the responding experts (18) from the total initially contacted (25) expressed their opinions on the positive factors, facilitators or enablers in relation to negative factors or barriers focusing on the Spanish health system and healthcare organizations, the nursing education at universities, the social recognition and image of nurses, and their interprofessional work.

It is worth noting in this first iteration the equality regarding the total number of responses by gender, being 9 women and 9 men. Finally, seven experts were excluded for not responding within the established time frame. The responses to the questions proposed were diverse but with many commonalities such as: the need to increase nursing training on NPM directly related with the current lack of NPM contents in the training in Pharmacology for nurses; the discrepancies with other health professionals about NPM; the rejection of some groups within the nursing workforce towards NPM; and the variability in the implementation and accreditation of prescribing nurses in the autonomous communities. The potential positive impacts, enablers or facilitators identified by the experts’ feedback as the most important strategies towards the completion of NPM in Spain were that by promoting NPM from a unified perspective, and with the collaboration of national health organizations, the higher education institutions, the public policy makers and the Boards and Councils of Nurses:the Spanish health system, which is highly collapsed due to its current organization, would be unblocked;the Spanish health system’s costs would be reduced;the higher education in pharmacology for nurses would be widened and improved with the creation of new research teams and initiatives;the specialization of nurses would be increased;the image and societal recognition of the Spanish nurses would be greater;the quality of care would become more holistic with patients-families-caregivers and would therefore be improved;the trust in the nursing workforce would increase;the interprofessional work and collaboration of health professionals and allied professions would be strengthened;the autonomy in nursing decision-making would be reinforced;the development of nursing skills towards advanced practice would increase;the generational renewal of nurses would be assured.

Moreover, the negative impacts and potential barriers highlighted by experts with regard to NPM implementation were:the problems within the nursing collective workforce, as some voices are against assuming new responsibilities such as NPM;the variability in the implementation nationwide and the lack of specific resources in the different autonomous communities;the issue of the term “prescription” and the euphemisms created, such as the name of the royal decree;the nursing profession’s ignorance about legal protection and therefore their fears towards putting it into practice;the ignorance and controversies about the criteria for the NPM accreditation;the uncertainty about the changes in the nurses’ salary regarding the implementation of NPM which implies a new professional responsibility;the lack of a common global framework in relation to the NPM aligned with Europe and the countries with good practices in NPM;the legal gaps and doubts in the Royal Decree Law;the lack of experience of the nurses;the lack of joint training designed for such purposes;the lack of professional confidence due to the lack of knowledge in the matter, which also affects patient safety;the distrust and low social visibility regarding nursing prescription;the ineffective or non-existing clinical performance;the lack of collaboration among those organizations and institutions involved.

### 3.2. Second Iteration

In Round Two, the responses of the experts (15) were related to the implementation of the NPM in Spain, as well as to the approved protocols and clinical practice guidelines for the specific medications. Seven responses were obtained from women and eight responses were obtained from men due to the exclusion of three experts for not sending their responses in the specified time frame. There were similarities in the points of view provided by the experts who belonged to the groups of Chief Nurse Officer (CNO), nurse supervisors, and nurses in clinical practice, and the answers provided by the professors, educators and academic managers of the different universities and also the representatives of the boards and councils of nursing with regard to the implementation of NPM.

All participants stated that it would be interesting to continue expanding NPM through more protocols and potential situations, especially in relation to the population with chronic illness, until a completely independent prescription or at least more prescriptive privileges for nurses could be achieved. They also stressed that this would make the public and private health system more agile and efficient, as the experience of other countries has demonstrated; and they also pointed out the need for a unified and consensual coalition, at least at the European level. In their responses, participants had ideas in common in relation to the fact that the implementation may cause certain interprofessional conflicts with other professionals such as physicians and pharmacists, by using the term “prescription” and they stated that by including a multidisciplinary PC model where the NPM could have independent functions and roles, the population health and health systems’ functioning will benefit.

However, regarding the role of the boards and councils of nursing on this topic, there were controversial responses from the experts, as they commented that those organizations have a fundamental role in professional decision-making with regard to the accreditation and development of the prescribing nurse, although the contributions those organizations have made until now in some regions have been scarce or practically non-existent.

### 3.3. Third Iteration

Finally, in Round Three the responses (15) were coincident with the previous iterations, and therefore the researchers believed that data saturation had been achieved and more rounds were not required. For this last round all the experts answered, with seven responses coming from women and eight responses coming from men. However, in this last iteration, various responses were obtained regarding the protocols or clinical practice guidelines where there was a consensus on the part of those currently in clinical practice or who have practiced for years in clinical settings and now had positions such as managers, academics, professors, or mixed profiles, reaching the conclusion that by promoting NPM, it is of vital importance to also develop new effective, safe and quality protocols as well as continuing professional development on NPM, either through specific postgraduate programs and Master degrees at universities, or continuous training organized by boards, councils of nurses, higher education institutions or even the health organizations.

The experts pointed out that other clinical situations should be addressed whereby nurses could prescribe medications, as well as the problem of the variability in the implementation speed within all of the autonomous communities and the setbacks in the publication of clinical practice guidelines by the Ministry of Health. In addition, the need for new clinical guidelines and protocols for the prescription of drugs in the treatment and follow-up of chronic pain, HIV and palliative care patients was found to be a common theme among the experts, allowing nurses to treat acute or chronic patients, and in turn, in different situations, such as primary care, specialized care at hospitals and home cares.

As regards to the section on nursing training in the field of medication prescription, the experts discussed the importance of enhancing the study programs at the university undergraduate level, specifically adding NPM contents and clinical cases in the course of Pharmacology, or even by adding a Nurse Prescribing specific course to the curriculum. Undoubtedly, the option with the highest response rate from experts regarding training corresponds to the support of initiatives of continuing professional education (CPD) throughout higher education institutions, boards and councils of nurses and health organizations that should be updated after a period of time.

## 4. Discussion

Studies from New Zealand and the United States of America have clearly shown that training for nurses in Pharmacology is of vital importance, leading to an increase in knowledge and confidence in the interprofessional team [19] and, consequently, increasing patient safety in the administration of medication by the nurse [20]. In addition, in the United Kingdom, licensed prescribing nurses have demonstrated that obtaining specific training on prescribing is essential, so they do not only focus on prescribing medications, but also on prescribing cares [21]. In line with these publications, our study provides evidences according to experts’ views that support the idea of increasing nursing training in Pharmacology under the umbrella of PC and focusing on NPM, as well as the provision of periodic updating to achieve an optimal level of knowledge in NPM.

The lack of proactive attitudes regarding interprofessional collaboration with other health professionals, such as physicians and pharmacists, in PC has a negative impact on the health of the patient, and these links are therefore essential to promote efficient and quality nursing care according to the European Federation of Nurses Associations [22]. Accordingly, in our study all experts stressed the importance of such interprofessional collaboration and also the collaboration among key stakeholders towards promoting NPM, what is also coincident with the statements and recommendations of the Nuphac Network and Demophac Project. In addition, we also show the disconnection between autonomous communities within the same country when making joint decisions, which does not benefit the unification of criteria and reduces the speed of implementation of NPM, consequently generating differences in nursing workforce development within the same country.

Contrasting our results with other studies carried out at national level, we have found similarities with the study published in Spain after the approval of the Royal Decree in 2018. It was a qualitative study conducted throughout a grounded theory approach with the objective of understanding the views of the health professionals who are involved in PC and NPM, in which interprofessional collaboration and more training for nurses on NPM came up as part of the results [16]. However, we have discrepancies with another study conducted in Catalonia which stated that nurses are perfectly qualified to prescribe medication. Such study indicated the option of providing the nursing workforce with more information on legislation, types of medications and health products before implementing the prescription process [23]. This discrepancy may be explained as a matter of interpretation by the fact that Catalonia has implemented a form of NPM within the Primary Care Attention earlier than other regions in Spain, and also because the type of information they mentioned that nurses need is coincident with the contents that our study emphasizes that should be present in the nursing training. Therefore it is not really a discrepancy but a different interpretation of the qualitative data collected in different contexts.

Moreover, the statement highlighted in our study about the development of a collaborative framework that could unify and standardize criteria among the different autonomous communities in Spain is coincident with the European Nuphac Network statements, as well as European Commission-funded projects such as Demophac and Eupron among others, as it would allow the evaluation and understanding of the figure of the prescribing nurse in Spain and in Europe, through the enhancement of the competencies in the field of medication prescription, improvement in communication, and the relationship of the interprofessional team, and the identification of errors and benefits in the patient [24,25,26].

### 4.1. Limitations

Despite the fact that there are 17 autonomous communities in Spain, we were only able to select experts from seven of them where the implementation of the Royal Decree has been carried out, generating different perceptions and expectations in health professionals. As our aim was to prioritize the experts’ background and experience with the topic, it has thus not been possible to obtain experts from all of the regions. However, we are convinced that even though the current laws on NPM in Spain may have different implementation speeds across the country, the same professional fundamentals and assumptions are equally present across all the regions.

In relation to the number of participating experts involved in the study, it was originally higher though some of them freely decided to abandon the sample in the first two rounds. Therefore, valuable opinions of experts who had greater knowledge about the topic were lost though we assume it is part of the complexity of our qualitative approach that demands more time and dedication in the responses of the experts, and could be one of the reasons why they decided not to continue. On the other hand even though the experts belonged to different areas of expertise, we are conscious that there may exist other areas that have not been represented in our study and could be appropriate to include them in future approaches.

### 4.2. Future Directions

This study is part of a PhD research that aims to develop strategies and resources for the Spanish nursing workforce towards advancing and implementing NPM nationwide. The results have provided key points for advancing NPM regarding education, research, practice, management, and policy from the perspective of the key stakeholders considered as experts for our study aims, and have stressed the need of a common framework aligned with Europe and other countries with experience in this topic, as well as the development of competencies throughout specific training and the promotion of nursing autonomy. We hope that our results could be also useful and helpful for the support and implementation of NPM in other countries where this nursing role has not been fully implemented.

## 5. Conclusions

After conducting our Delphi study for the identification of the enablers and barriers to the implementation of NPM in Spain, the experts reached a consensus on the relevance of proposing improvements in the Spanish educational programs. These included the introduction of specific contents about NPM at universities and at different academic levels, stressing the specialization of nurses depending on the areas where they develop their professional activity, promoting the autonomy of nurses, providing greater competencies to the nursing workforce for decision-making at the national, regional and local level, and promoting the figure of the prescribing nurse towards higher levels of independency as well as the development of a unified framework on PC and NPM and new clinical practice guidelines or individualized protocols for more types of chronic diseases.

## Figures and Tables

**Figure 1 ijerph-20-04681-f001:**
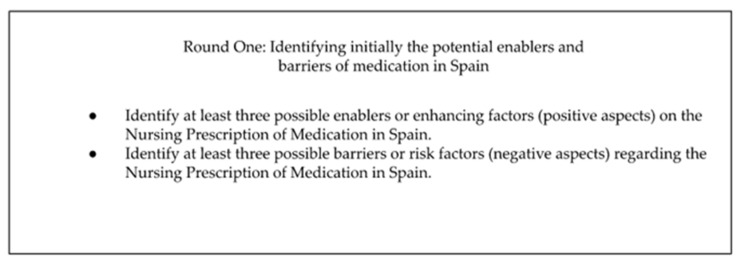
Round One: Identifying initially the potential enablers and barriers of medication in Spain.

**Figure 2 ijerph-20-04681-f002:**
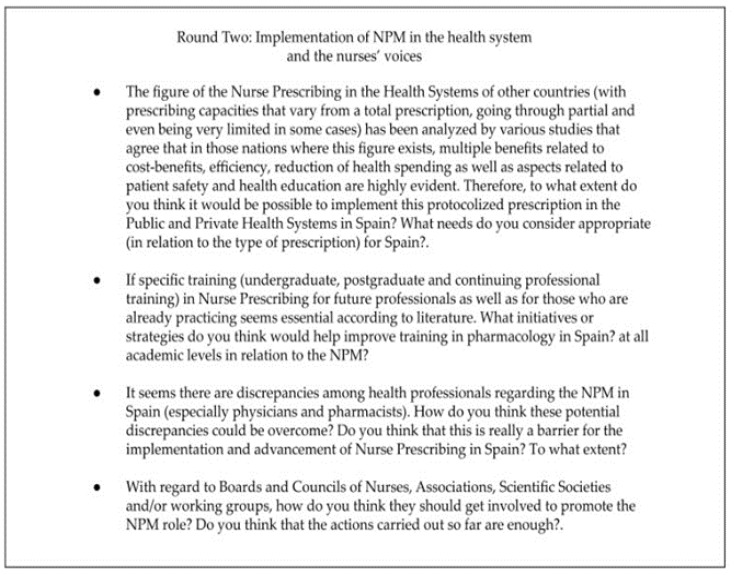
Round Two: Implementation of NPM in the health system and the nurses’ voices.

**Figure 3 ijerph-20-04681-f003:**
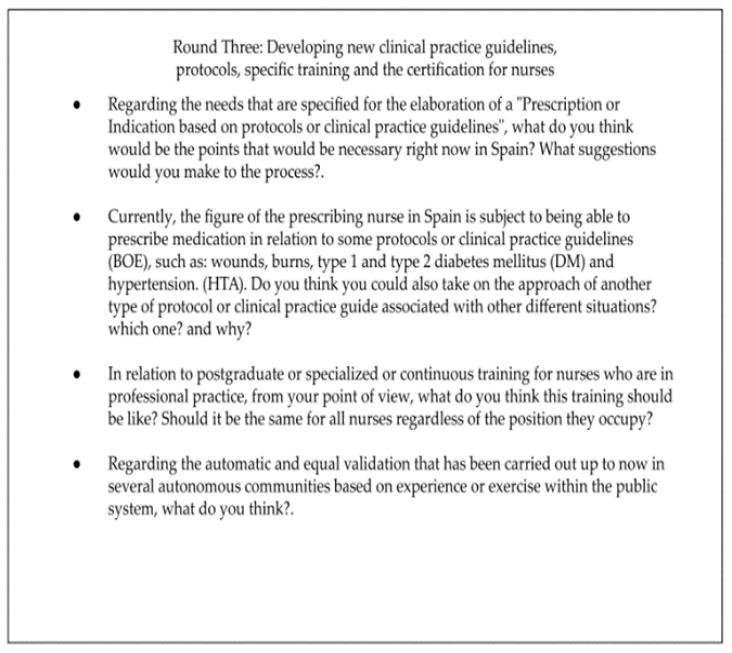
Round Three: Developing new clinical practice guidelines, protocols, specific trainings and the certification for nurses.

**Table 1 ijerph-20-04681-t001:** Sociodemographic characteristics of the experts.

Sociodemographic Characteristics of the Experts (n = 15)			N (%)
Gender			
Female			5 (34)
Male			10 (66)
**Autonomous Community**			
Valencian Community			7 (48)
Andalusia			2 (14)
Galicia			2 (14)
Castilla La-Mancha			1 (6)
Madrid			1 (6)
Basque Country			1 (6)
Catalonia			1 (6)

**Table 2 ijerph-20-04681-t002:** Professional and academic characteristics of the experts.

Professional and Academic Characteristics of the Experts (n = 15)			N (%)
**Professional Status**			
Working exclusively in clinical practice			
Clinical Nurse			3 (20)
Nurse Supervisor			1 (6)
CNO (Private Hospital)			1 (6)
Working exclusively in academic field			
Educator/Teacher/Professor			4 (28)
Academic Manager			1 (6)
Combined professional profiles			
CP and AF			2 (14)
CP and boards/CN			1 (6)
CP and AF and boards/CN			2 (14)
Years of Professional Experience			21
Academic Level			
Undergraduate			2 (13)
Postgraduate/Master/Degree			6 (43)
PhD			7 (46)

Abbreviations: AF = Academic Field; CNO = Chief Nurse Officer; CN = Council of Nurses; CP = Clinical Practice.

## Data Availability

Not applicable.

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
