# Peer review of "Identifying the Enablers and Barriers to Advance Nurse Prescribing of Medication in Spain According to Experts’ Views: A Delphi Study"

_ijerph, 2023, doi:10.3390/ijerph20064681_

Round 1

Reviewer 1 Report

In this paper, the authors conduct a Delphi study on the attitudes of various experts on implementing widespread prescriptive authority to nurses in Spain, A  Delphi consists of different phases of data collection and analysis following an iterative process towards reaching a consensus on a particular area, utilizing expert opinion.

The methodology and conclusions of the study are clear. 

The authors should set the stage for why it would be useful for nurses to have prescriptive authority by providing some background on current deficiencies of physician and advanced practice nursing personnel in Spain and Europe.

Most of my comments below are to improve the readability of the scientific English. 

Nurses play an important role in pharmaceutical care worldwide in detecting clinical changes, communicating and discussing pharmacotherapy with patients, their families, and other healthcare professionals, proposing and implementing drug-related interventions, and ensuring the monitoring of patients and their medication regimens, among others. However, there is not a global consensus about regarding nurse prescribing of medication worldwide and sometimes not even nationwide. In Spain, for example, this topic is currently in transition since the approval of the Royal Decree {EXPLAIN THE ROYAL DECREE IN THE ABSTRACT}

. Our study aims to identify the enablers and barriers to advancing nurse prescribing of medication in Spain through the views of experts in the field and according to the latest Royal Decree approved and the steps taken by the different Spanish Autonomous Communities. A modified qualitative Delphi study with three iterations was performed online throughout the views perspectives of experts from the field of healthcare education, research, practice, management, and policy. Data extracted from the literature were used to formulate the open-ended questions utilized in the 3 three rounds. The experts involved (n = 15) belonged to different Spanish regions where the Royal Decree is being implemented with different speeds and had distinct backgrounds and experiences. Our results highlight the importance of developing more protocols based on chronic diseases, scaling up towards independent prescription, the inclusion of a joint multidisciplinary pharmaceutical care model, the controversial role played by nursing councils and boards, the variability in the speed of implementation among all the Autonomous Communities and the lack of nursing training in the field of medication prescription.

Line 40: nurses should be small case

Line 47: “such a situation it is worth noting the initiatives of the Nuphac Network aiming at indicating the functions of nurses in interprofessional pharmaceutical care, showing the ideal roles of nursing and, in turn, creating the opportunity to translate them into study pro- 49 grams oriented to said competencies and adjusted to the expectations of the labor market”

Better: such a situation it is worth noting the initiatives of the Nuphac Network aiming at indicating delineating the functions of nurses in interprofessional pharmaceutical care, showing the ideal roles of nursing and, in turn, creating the opportunity to translate them into study pro-grams oriented to said competencies and adjusted to the expectations of the labor market

Line 53: Better: Pan-European model on of PC and having

Line 57: use semi-colon instead of comma; remove “for sure”

Better: providing patient education and information; monitoring medication adherence, adverse and therapeutic effects; and for sure prescribing medicines

Line 59: Better: The regulations on NPM in Spain have has evolved since the first Law 44/2003 of November 21st  was created referring to the prescription provision of cares care and comprising comprehensive health care services, assuming cooperation, a multidisciplinary approach, the integration of processes and care continuity, and avoiding the division between the care processes attended performed by different specialists.

Line 78: the Andalusian Public Health System to use and indicate recommend medicines not

Line 99:  Better: In the 4-year undergraduate nursing program at Spanish universities, a course in pharmacology is generally taught during the second academic year.

Line 118: better: The language used for any all communications in our study was Spanish. At Line 119: round the principal researcher investigator explained through email the process to be followed by the participants and in addition a word a document was ...

Line 123: participant privacy and confidentiality.

Line 131  … only 15 experts arrived continued until the end …

Line 139: better: After deciding on participation, they provided informed consent.

Line 144: better: Each expert did not know the identity of the others on the panel.

Line 145: In addition, with Round Three, the sociodemographic, professional, and academic characteristics of each expert were collected (see Tables 1 and 2).

Line 161: nurses’ voices perspectives

Line 164:  and other nursing professional parties organizations..

Line 170 From the results of Round Two, many of the topics related to NPM in Spain were obtained found to be useful for generating the third questionnaire,

Line 175: In addition, a file questionnaire was also attached

Line 176: Ssociodemographic Data profile such as: … regarding their areas of expertise, …

Line 193: In Round 1 all the responding experts (18) from the total firstly first contacted

196 nursing education at universities, The the social recognition and image of nurses, 197 as well as their

Line 200:  Finally, 7 experts were excluded for not responding within the established specified time frame.  

Line 203:  the nursing training in this topic liked linked with the current lack of training in pharmacology

Line 206: accreditation of prescribing nurses in the autonomous communities. 207

Line 215• the nursing higher education in pharmacology would be widened and improved

Line 216: with the creation of new research trends teams and initiatives;

218: recognition of the Spanish nurses

Line: 223 allied professions would be assured strengthened

Line 244: the lack of professional security confidence due to …

Line 255: in the established specified time frame.

Line 261: chronic patients with chronic illness until a completely independent prescriptive privileges could be achieved.

Line 271: boards and councils of nursing

Line 294: … follow-up of chronic pain, HIV, and palliative care patients was obtained as a common theme

Table 2: Professional Characteristics of the Experts

Line 368: although there was a mortality an attrition of the sample in the first two rounds

Author Response

Dear Reviewer,

Many thanks for your time, effort and expertise in revising our manuscript.

We included all your comments and suggestions.

Grettings

Reviewer 2 Report

The study aims to identify the enablers and barriers to advance nurse prescribing of medication in Spain through a modified qualitative Delphi study in three iterations.

-          Are there any considerations why the quantitative method was not applied?

-          Why did the known enablers and barriers from the previous studies (e.g. determined through literature review) not identified first and shown to the expert and asked them to rate it? If the respondents have other thoughts, they can fill in the description. By doing so the agreement among respondents could be assessed objectively through quantitative methods.  

Author Response

Dear Reviewer,

Many thanks for your time, effort and expertise in revising our manuscript.

We appreciate your comments and suggestions.

Please see the attached file to read the answers to your questions.

Grettings

Reviewer 3 Report

An adequate presented qualitative research paper although perhaps too specific for Spanish own characteristics of regions composition and health system delivery. 

On methodology there is a need to know about composition of the core group that performed the study. Also information about the way that the experts were chosen for the Delphi process as there might be a selection bias. Although qualitative studies permit not large number of participants on this occasion perhaps numbers were not big enough for lack of some regions representation. There is a lack of information about exact background of some of the responders on the academic field ( any medical professional at all?) Perhaps also the need for more participants would be necessary to achieve more practice representation as might be necessary to better balance practical aspects on nurse prescription.

Author Response

(The authors gave the same response as above.)
